# Wood-Decay Fungi Fructifying in Mediterranean Deciduous Oak Forests: A Community Composition, Richness and Productivity Study

**Ángel Ponce** [1,2]**, Elena Salerni** [3,]*****, Maria Nives D'Aguanno** [4] **and Claudia Perini** [3,5]

1 Department of Agricultural and Forest Sciences and Engineering, University of Lleida,
  Av. Alcalde Rovira Roure 191, E-25198 Lleida, Spain; angel.ponce@udl.cat
2 Joint Research Unit CTFC—AGROTECNIO—CERCA, Av. Alcalde Rovira Roure 191, E-25198 Lleida, Spain
3 Department of Life Sciences, University of Siena, Via P.A. Mattioli 4, 53100 Siena, Italy; claudia.perini@unisi.it
4 Institute of Microbiome and Applied Sciences, Malta Life Sciences Park (LS2.1.12—LS2.1.15),
  SGN 3000 San Gwann, Malta; mariadaguanno1@gmail.com
5 NBFC, National Biodiversity Future Center, 90133 Palermo, Italy
* Correspondence: elena.salerni@unisi.it

**Abstract:** Deadwood is involved in several important ecological roles, being the fundamental habitat of wood-decay fungi. At the same time, this polyphyletic group of fungi is the principal agent of wood decomposition, regulating the carbon cycle and the food resource for many other organisms. It is known that the diversity and community composition of wood-decay fungi are related to the tree species, decay stage and size of the woody debris in which they are living. Nevertheless, there is a lack of information on Mediterranean *Quercus cerris* L. forests. In response, we explored how wood-decay fungi colonize different woody types and how the productivity, richness and community composition of these fungi is influenced by the decay stage and size of the deadwood. Our results indicate that the studied groups, i.e., Ascomycetes, Corticioids, Polyporoids and Heterobasidiomycetes responded differently to the woody debris classes. Moreover, we note the high importance of smaller and soft-decayed woody debris for community composition and richness, hosting a great number of species, in addition to the positive effect of the heterogeneity of the woody debris size for wood-decay fungi productivity.

**Keywords:** wood-inhabiting fungi; woody debris; decay stage; size; *Quercus cerris*; State Nature Reserves; central Italy

## 1. Introduction

Deadwood is a vital component of forest ecosystems, fulfilling several important ecological roles. It plays a key role in carbon storage and the regulation of nutrient cycling [1–4], as well as in hydrologic processes [5,6]. Moreover, deadwood serves as a habitat for numerous organisms [1,7,8]. In fact, a variety of fungi, plants and animals have different degrees of association with deadwood, providing suitable microhabitats for growth, reproduction, shelter and nutrient sources [9].

Wood-decay fungi are particularly significant among organisms depending on deadwood [10,11]. These ecologically and functionally important organisms are the principal agents of wood decomposition, regulating the carbon cycle and food resources for many other groups [12]. Wood-decay fungi encompass a polyphyletic group with diverse life forms, including Agaricoids, Corticioids, Polypores, Hydnums, jelly and coralloid fungi [13], degrading differently the dead wood resource [14]. In fact, the relationship between wood-decay fungi and the deadwood resource is highly intimate, and the diversity and community composition of these fungi are primarily influenced by the size of the woody debris [2,13,15].

Despite the high importance of wood-decay fungi in forest ecosystems, their communities within Mediterranean areas have received limited attention in previous studies [16–20]. Moreover, Mediterranean areas are recognized as crucial biodiversity hot spots, harboring a wide range of animals and plant species [21]. Recent research, highlighted in a specialized publication, has also emphasized the importance of these areas for a wide range of fungal species [22], being closely related to the vegetation and tree species on which they grow [23]. Furthermore, the decay stage and size of the debris also influence the abundance and composition of wood-decay fungi [16,18]. While previous research has predominantly focused on larger woody debris classes [9,17], studying smaller size pieces is crucial due to their ecological value [13,15]. However, studies on the productivity of wood-decay fungi remain scarce, as most of the previous research has primarily examined richness and community composition [16,24].

In summary, previous research has highlighted the significance of host tree species and the size of dead wood in shaping fungal diversity [25,26]. However, there has been limited investigation into how dead wood also impacts wood-decay fungi productivity. Therefore, the aim of this study is to examine the wood-decay fungal community growing in Mediterranean deciduous oak (*Quercus cerris* L.) forests by (i) describing how wood-decay fungi colonized different woody debris features (ii) analyzing how the productivity, richness and community composition are influenced by the size of the deadwood.

## 2. Materials and Methods

### 2.1. Study Area

The study area is located within the two State Nature Reserves of Cornocchia (43°23' N, 11°10' E) and Palazzo (43°20' N, 11°04' E) in the province of Siena, Tuscany, central Italy. These reserves encompass approximately 800 ha of meadows and pastures situated on hillsides with varying slopes (ranging from 5% to 42%) and altitudes (ranging from 285 to 531 m.a.s.l). Both study areas share similarities in terms of bedrock, with limestone, sandstone and siltstone being present. The soils in these areas are near-neutral, and the forest type, composition, and density are also comparable. It is worth noting that no logging or harvesting activities have occurred in either of the reserves for the past four decades [27]. The stands are composed of a dominating *Quercus cerris* L. in the canopy layer, followed by *Franixus ornus* L. and *Q. pubescens* Willd., covering from 55 to 90%. The number of trees with a diameter at breast height, (DBH) > 2 cm, ranged from 7 to 33 trees in 1000 m$^2$, with a mean tree density of $17 \pm 7$ per 100 m$^2$. Secondly, shrubs such as *Acer campestre* L., *Cornus mas* L., *Crataegus monogyna* Jacq., *Juniperus communis* L. and *Ruscus aculeatus* L., covered from 3 to 70% of the study area, while herbaceous plants such as *Anemone nemorosa* (L.) Holub, *Brachypodium rupestre* (Host) Roem. & Schult., *B. sylvaticum* (Huds.) P. Beauv., *Carex flacca* Schreb., *Viola alba* Besser and *Prunella vulgaris* L. covered from 1 to the 75%. The soils are mostly blanketed by dead biomass from the adjacent trees, i.e., litter, branches and logs, covering from 50 to 95% of the forest surface. The climate is characterized by a dry summer and rain in spring and autumn; the hottest months are July–August with 22 °C and the coldest are January–February with a mean of 6 °C [28,29]. The mean annual precipitation is approximately 800 mm, and the mean annual temperature is 13.5 °C [27].

### 2.2. Sampling Design

Twenty-four 100 m$^2$ permanent plots (10 × 10 m$^2$) marked by metal stakes in each corner were randomly placed in the two Natural Reserves. The plots were previously identified and mapped (scale 1:5000) by photointerpretation, with a buffer zone of about 20 m around each polygon to reduce possible edge effects. Each plot was surveyed every May, October, November and December from the autumn of 2012 until the spring of 2014 to collect all wood-decay fungi from all woody debris. All sporocarps belonging to the same taxon found on a single piece of woody debris were considered a single occurrence. Fungal samples were collected and subsequently dried for later microscopic identification at the species level whenever possible. Most of the species were identified in the Laboratory of

Mycology, Department of Life Sciences, University of Siena, and were studied by the usual macro- and micromorphological techniques using analytical keys [30–36]. The studied groups were Ascomycetes, including stromatic Sordariomycetes (Ascomycetes with a large stroma, mainly Xylariales) and Discomycetes; Corticioids fungi, Homobasidiomycetes with smooth to odontoid hymenophore on a soft, resupinate sporocarp; Polyporoids, Homobasidiomycetes with a hymenophore in the shape of tubes on the underside of the sporocarp; Heterobasidiomycetes, basidiomycetes with gelatinous sporocarp, i.e., 'jelly fungi'. The nomenclature of the species refers to the 'CABI-Bioscience Database of Fungal Names', updated to April 2023 [37] and the exsiccatum was preserved at the *Fungarium* section of the *Herbarium Universitatis Senensis* (SIENA).

During the fieldwork, detailed information regarding the diameter and decay stage of the associated woody debris pieces was recorded for each fungal specimen. Only a few pieces of debris were identified at a specific level due to obvious difficulties in recognizing fine, decomposed branches. The woody debris was categorized into specific diameter classes based on the classification system established by Küffer and Senn-Irlet [38] and Abrego and Salcedo [19]. These categories include:

- Very Fine Woody Debris (VFWD): comprising branches and twigs with a diameter of $\leq$ 5 cm.
- Fine Woody Debris (FWD): encompassing logs with a diameter between 5 and 10 cm.
- Coarse Woody Debris (CWD): including logs or snags with a diameter of $\geq$ 10 cm.
- Stumps: for the wood-decay fungi preferences for species-specific woody debris sizes and decay description (surveyed during 2012, 2013 and 2014), stumps are categorized as CWD according to the classification by Albrecht et al. [39].

Additionally, the decay stages of the dead wood pieces were assessed using a knife. The following decay stages were identified for each woody debris piece:

- Decay Stage 1 (DS1): Characterized by recently dead or cut trunks or pieces of wood, with firm wood, fresh bark and phloem. In this stage, the knife only penetrates a few millimeters into the wood.
- Decay Stage 2 (DS2): Represents an intermediate decomposition stage, where the wood is partially decayed and often accompanied by loosened pieces of bark. At this stage, the knife typically penetrates $2 \pm 5$ cm into the wood.
- Decay Stage 3 (DS3): Corresponds to the advanced stages of decomposition, where most of the wood is soft throughout. In this stage, the entire blade of the knife easily penetrates the wood.

### 2.3. Plot-Specific Woody Debris Variables

In order to study how the productivity, richness and community composition of wood-decay fungi are influenced by the plot-specific size of the woody debris, we considered the abundance and variety (from here onwards, Variety) of the wood surveyed in each forest plot in 2013, in addition to the wood-decay fungi found during the same year (Table 1). The abundance of woody debris is defined by counting the amount of debris in each plot for different size classes, including VFWD, FWD, CWD and stumps. Total abundance is defined as the sum of all the woody debris of all classes per plot. According to Abrego and Salcedo [18], Variety was measured by the Simpson's Diversity Index. It is a measure of diversity, which considers both the number of woody debris and their evenness, providing a measure of the probability that two randomly selected pieces from an area would belong to the same woody debris class.

**Table 1.** Summary of the plot-specific modeling data.

|  | Min. | Mean | Median | Max. |
|---|---|---|---|---|
| Total sporocarp productivity | 8 | 26 ± 9.43 | 25 | 42 |
| Ascomycota sporocarp productivity | 0 | 1 ± 1.47 | 1 | 5 |
| Corticioids sporocarp productivity | 4 | 20 ± 7.93 | 17 | 35 |
| Polyporoids sporocarp productivity | 0 | 3 ± 2.73 | 3 | 10 |
| Heterobasidiomycetes sporocarp productivity | 0 | 1 ± 1.40 | 1 | 5 |
| Total sporocarp richness | 8 | 14 ± 4.37 | 15 | 23 |
| Ascomycota sporocarp richness | 0 | 1 ± 1.14 | 1 | 4 |
| Corticioids sporocarp richness | 4 | 9 ± 3.07 | 9 | 16 |
| Polyporoids sporocarp richness | 0 | 3 ± 1.97 | 2 | 7 |
| Heterobasidiomycetes sporocarp richness | 0 | 1 ± 0.86 | 1 | 3 |
| VFWD abundance | 15 | 35 ± 13.64 | 32 | 72 |
| FWD abundance | 0 | 6 ± 4.68 | 6 | 19 |
| CWD abundance | 0 | 2 ± 1.39 | 2 | 5 |
| Stump abundance | 0 | 3 ± 2.00 | 2 | 8 |
| Total woody debris | 23 | 46 ± 16.64 | 41 | 85 |
| Variety | 0 | 0 ± 0.13 | 0 | 1 |

Abbreviations: VFWD: Very fine woody debris; FWD: Fine woody debris; CWD: Coarse woody debris.

### 2.4. Data Analysis

In order to describe the wood-decay fungi preferences for species-specific woody debris sizes and decay stages surveyed from 2012 to 2014, we plotted the groups found on the plots in addition to the most abundant species. Here, we define the most abundant species as the ones that were observed more than 95 times during the whole study period. Secondly, we constructed Venn diagrams to visualize the distribution of wood-decay fungi across different decay stages and woody debris diameters.

For studying how the plot-specific wood-decay fungal productivity and richness are influenced by the size of the woody debris we used general linear models (GLM). In these models, we used sporocarp productivity and sporocarp richness as response variables. Specifying, sporocarp productivity was defined as the sum of sporocarp counts per plot, while sporocarp richness was defined as the sum of sporocarp species number per plot. On the other hand, the wood variables were assigned as explanatory variables. Wood-decay fungal productivity and species richness were analysed separately for total Ascomycetes, Corticioids, Polyporoids and Heterobasidiomycetes groups. To select the best model, we carefully considered the ecological relevance of the variables included in the analysis, secondly, Akaike's information criterion (AIC), and the absence of collinearity (Variance Inflation Factor < 3) between variables, using the 'performance' package [40].

To analyze the sporocarp community composition across plots, we conducted a detrended correspondence analysis (DCA) using the sporocarp matrix of the species. Only the species that reached the 10th percentile of the sum of sporocarp abundances were included in this analysis to reduce noise effects. Additionally, we investigated whether these spatial changes in community composition could be explained by the deadwood variables using the passive fit over the previous DCA ordination. All tests were run in five independent response data sets: (i) sporocarps for the total community composition, (ii) sporocarps for the Ascomycetes community composition, (iii) sporocarps for the Corticioids community composition, (iv) sporocarps for the Polyporoids community composition, (v) sporocarps for the Heterobasidiomycetes community composition in order to inquire in possible differences between the different wood-decay fungi groups.

The data analyses were implemented and carried out in R software 4.0.2 [41] using the 'glm' function implemented in the 'stats' package and part of R, 'vegan' package for multivariate analyses [42].

## 3. Results

### 3.1. Wood-Decay Fungi Community Composition

Over the course of four consecutive spring and autumn fruiting seasons, we documented a total of 1497 sporocarps. Among these, there were 346 spring sporocarps, with Corticioids comprising 80% of this season's sporocarps. Ascomycetes and Polyporoids accounted for nearly 8% each, while Heterobasidiomycetes represented 4% of the spring sporocarps. In the autumn season, we collected a total of 1151 sporocarps, with Corticoids contributing 75%, Polyporoids making up nearly 15%, Ascomycetes accounting for 6%, and Heterobasidiomycetes contributing 4% of the total. When studying the wood-decay fungi richness, we determined a total of 156 taxa at the species level and 6 at the genus level, divided into 75 species for spring and 143 for autumn. The distribution of species was similar to the productivity results, with 86 Corticioids, 36 Polyporoids, 20 Ascomycetes and 14 Heterobasidiomycetes. A comprehensive list of all the identified species and taxa can be found in Table A1 in Appendix A.

When plotting the woody debris preferences for the groups studied, we did not detect a specific preference for the different decay stages and sizes of the woody debris. Although, Ascomycetes and Heterobasidiomycetes were found mostly in soft decayed woody debris (DS1). In contrast, Corticioids appeared to be more flexible, being associated with a wider range of woody decay stages and sizes. Polyporoids, on the other hand, exhibited a more generalist behavior, being capable of growing across various decay stages and woody debris sizes (Figure 1A). Regarding the most abundant species, all of them belonged to the Corticioid group and had different requirements when fructifying on the woody debris (Figure 1B). *Peniophora quercina* and *Vuilleminia comedens* displayed a preference for thinner woody debris sizes and were only found on less decayed woody debris. *Stereum hirsutum*, on the other hand, demonstrated a more generalist behavior but still showed a preference for less decayed woody debris. Moreover, *Schizopora paradoxa* exhibited a preference for smaller woody debris sizes (i.e., VFWD and FWD) but could be found in different decay stages (Figure 1B). Other species found to a lesser extent can be cited, such as *Ceriporia purpurea*, growing on very rotten wood (DS1 and DS2) as already mentioned by [43], and *Stereum reflexulum*, a rare Mediterranean species fructifying on VFWD [44,45].

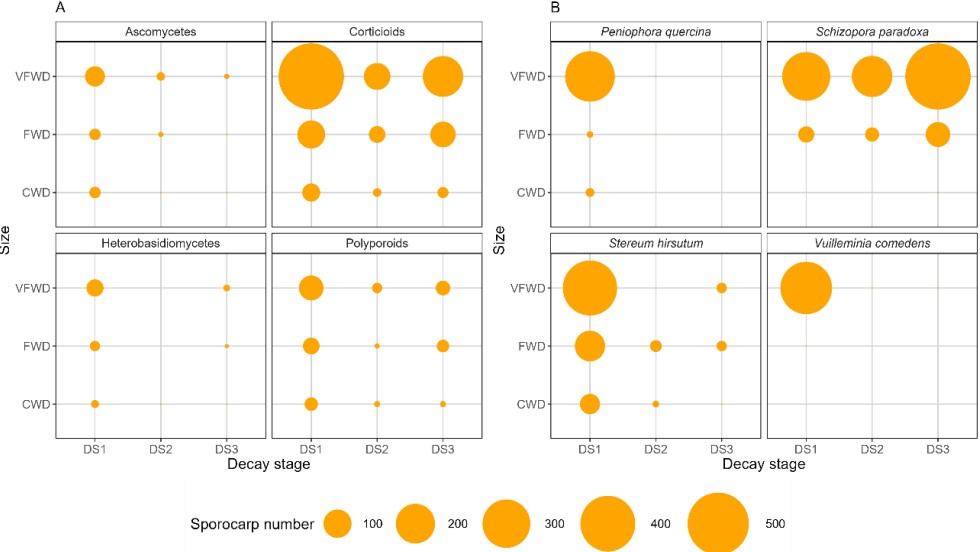

**Figure 1.** Wood-decay fungi preferences for the different woody debris types. Preferences graphs are built for each group (**A**) and most abundant species (**B**) studied while yellow circles indicate the combination of the decay stage and woody size in which the wood-decay fungi were found. *X* axis represents the decay stages: DS1, the first decay stage; DS2, the second decay stage; DS3, the third decay stage. *Y* axis represents the woody size studied: VFWD, very fine woody debris; FWD, fine woody debris; CWD, coarse woody debris.

The Venn diagram reported higher richness values for DS1 with 63 species, then for DS3 with 20 species and, lastly, for DS2 with 5 species. In addition, more than 45% of the species were found in more than one decay stage, usually in both DS1 and DS3 decay stages (Figure 2A). For the woody size, and similar to the decay stages, more than 45% of the species were found growing in more than one woody size typology. Moreover, VFWD hosted 64 species growing exclusively in this size class, followed by FWD with 20 species and CWD with 6 species (Figure 2B).

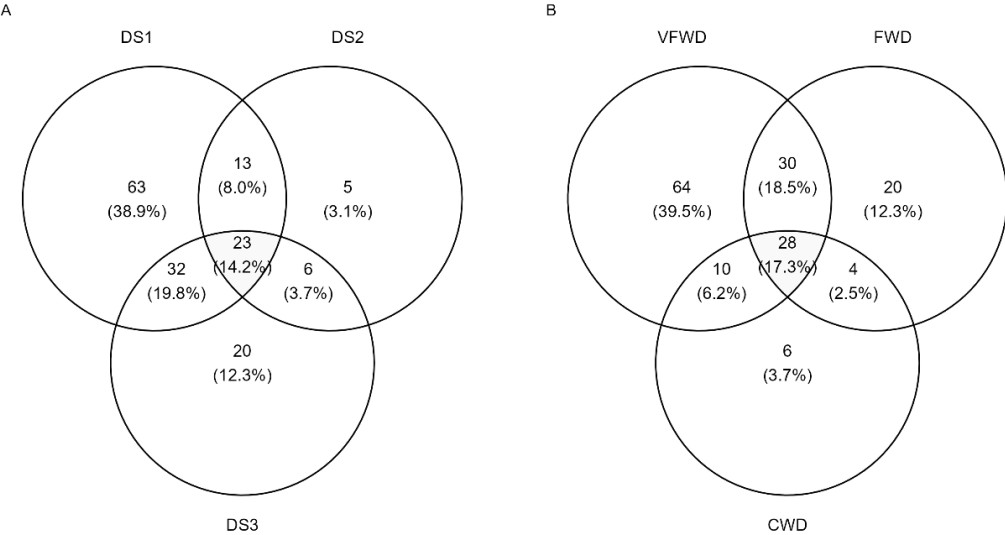

**Figure 2.** Venn diagrams of species diversity. In (**A**), circles indicate the different decay stage types analysed: DS1, first decay stage; DS2, second decay stage; and DS3, third decay stage. In (**B**), circles indicate the different woody sizes: VFWD, very fine woody debris; FWD, fine woody debris and CWD, coarse woody debris. Numbers inside the circles indicate the number of species while overlapping areas demonstrate the shared number of species.

### 3.2. Wood-Decay Fungi Productivity and Diversity and Plot-Specific Woody Size

When analyzing the effect of the woody debris variables on the sporocarp productivity, we saw that total sporocarp productivity was positively correlated with Variety of the woody debris (Table 2). Secondly, Ascomycetes productivity was significantly and positively correlated with the number of VFWD found on the plots. When checking the Corticioids group, we saw that there is a significant and positive correlation with Variety of woody debris. We found a significant and positive correlation with the number of FWD and CWD for the Polypores productivity. Finally, when testing the variables that influenced the Heterobasidiomycetes productivity we did not detect significant variables (Table 2).

Testing the effect of the woody debris on the species richness, we saw that total sporocarp productivity was significantly and positively correlated with Variety of woody debris. Following, Ascomycetes was significantly and positively correlated with the number of VFWD. Corticioids and Polypores groups were significantly and positively correlated with Variety of woody debris. For Heterobasidiomycetes, we did not find any significant variable explaining its richness (Table 2).

**Table 2.** Statistically significant parameter estimates for the selected models. The response variables tested are sporocarp productivity and sporocarp richness divided into total Ascomycetes (Asco), Corticioids (Corti), Polypores (Polyp) and Heterobasidiomycetes (Heterob). Significance levels: *p*-value < 0.001 '***', *p*-value < 0.01 '**', *p*-value < 0.05 '*'.

| Explanatory Variables | Model | | | | | | | | | |
|---|---|---|---|---|---|---|---|---|---|---|
| | Sporocarp Productivity | | | | | Sporocarp Richness | | | | |
| | Total | Asco | Corti | Polyp | Heterob | Total | Asco | Corti | Polyp | Heterob |
| Intercept | +2.60 *** | −0.75 | +2.41 *** | +0.54 * | - | +2.01 *** | −0.88 | +1.68 *** | −0.09 | - |
| VFWD | - | +0.03 ** | - | - | - | - | +0.03 * | - | - | - |
| FWD | - | - | - | +0.06 ** | - | - | - | - | - | - |
| CWD | - | - | - | +0.28 ** | - | - | - | - | - | - |
| STUMP | - | - | - | - | - | - | - | - | - | - |
| Total_WD | - | - | - | - | - | - | - | - | - | - |
| Variety | +1.70 *** | - | +1.43 *** | - | - | +1.66 *** | - | +1.32 * | +2.74 ** | - |

Abbreviation: VFWD, Very fine Woody debris; FWD of Fine Woody debris; CWD, Coarse Woody debris; Total_WD, sum of all the woody debris sizes.

### 3.3. Wood-Decay Fungi Community Composition and Plot-Specific Woody Size

The DCA ordination for the total community composition produced eigenvalues (λ) of 0356, 0.264, 0.222 and 0.179, and GL of 2.416, 2.608, 2.022 and 2.198 for the first four axes, respectively. The ordination showed that sporocarp composition was similar between plots since all plots are mostly in the central area of the ordination (Figure 3A). The community was principally driven by the differences between plots as shown in DCA1, although we did not see a significance of the Plot variable (explained variation = 3%, *p* = 0.760). DCA2 was separated on the negative end by plots with a higher amount of fine and coarse woody debris (FWD, explained variation = 25%, *p* = 0.048; CWD, explained variation 44%, *p* = 0.002) while the positive end of axis 2 had a lower number of these two woody sizes (Figure 3A). The relative abundances of species such as *Cyanosporus subcaesius* (Cyasub) and *Hypoxylon rubiginosum* (Hyprub) were related to higher amounts of CWD and species such as *Xylaria hypoxylon* (Xylhyp) and *Xylodon raduloides* (Xylrad) were more present in plots with higher FWD amount (Figure 3B).

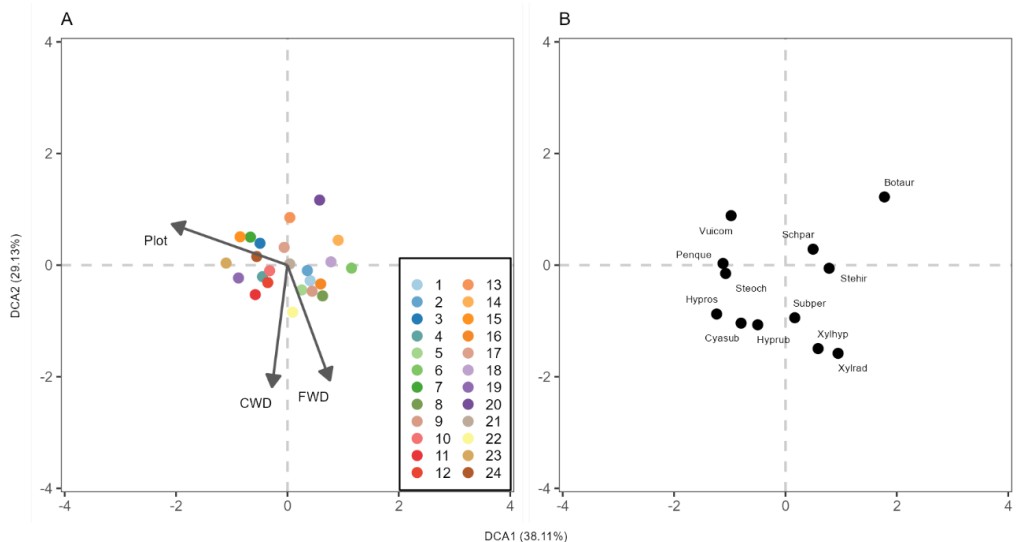

**Figure 3.** Detrended correspondence analysis (DCA) of total wood-decay fungi sporocarp community composition across plots. In (**A**), the studied plots are indicated with different dot colors while both Plot variable and the significant deadwood variables are indicated with arrows. In (**B**), the most abundant species are represented. The percentage of variance explained by each DCA axis is in

parentheses. Abbreviations of fungal species shown in the DCA diagram: Botaur—*Botryobasidium aureum*, Cyasub—*Cyanosporus subcaesius*, Hypros—*Hyphoderma roseocremeum*, Hyprub—*Hypoxylon rubiginosum*, Penque—*Peniophora quercina*, Schpar—*Schizopora paradoxa*, Stehir—*Stereum hirsutum*, Steoch—*Steccherinum ochraceum*, Subper—*Subulicystidium perlongisporum*, Vuicom—*Vuilleminia comedens*, Xylhyp—*Xylaria hypoxylon*, Xylrad—*Xylodon radula*.

The DCA ordination for the Ascomycota community composition produced eigenvalues (λ) of 1.000, 0.537, 0.2004 and 0.103, and GL of 1.033, 2.809, 1.421 and 0.660 for the first four axes, respectively. Plot (explained variation = 2%, $p = 0.783$) and FWD (explained variation 37%, $p = 0.017$) were the variables that separated the community in the DCA1, although no significant effect was found in the Plot variable (Figure 4A). No variables explaining the distribution of the community composition along the DCA2 were detected.

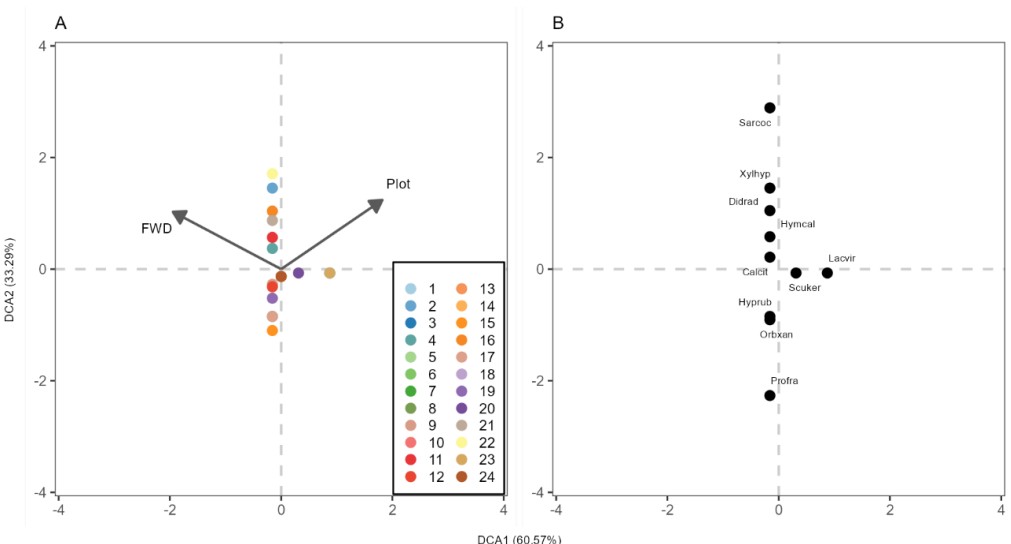

**Figure 4.** Detrended correspondence analysis (DCA) of Ascomycota sporocarp community composition across plots. In (**A**), the studied plots are indicated with different dot colors while both Plot variable and the significant deadwood variables are indicated with arrows. In (**B**), the most abundant species are represented. The percentage of variance explained by each DCA axis is in parentheses. Abbreviations of fungal species shown in the DCA diagram: Calcit—*Calycina*, Didrad—*Diderma radiatum*, Hymcal—*Hymenoscyphus calyculus*, Hyprub—*Hypoxylon rubiginosum*, Lacvir—*Lachnum virgineum*, Orbxan—*Orbilia xanthostigma*, Profra—*Propolis farinosa*, Sarcoc—*Sarcoscypha coccinea*, Scuker—*Scutellinia kerguelensis*, Xylhyp—*Xylaria hypoxylon*.

Secondly, the DCA ordination for the Corticioids community composition produced eigenvalues (λ) of 0.253, 0.241, 0.168 and 0.121, and GL of 2.482, 2.518, 1.883 and 1.8493 for the first four axes, respectively. We did not find variables explaining how species and plots were distributed among DCA1 while DCA2 was separated at the positive end by Plot (explained variation = 5%, $p = 0.606$) and by CWD (explained variation = 43%, $p = 0.002$) (Figure 5A). The relative abundance of species such as *Phlebia subochracea* (Phlsub), *Peniophora lycii* (Penlyc) and *Byssomerulius corium* (Byscor) seemed to be related to plots with higher amounts of CWD while species belonging to the genus *Lyomyces* such as *L. juniperi* (Lyojun) and *L. pruni* (Lyopru) were more related to plots with lower amounts of this kind of debris (Figure 5B). The DCA ordination for the Heterobasidiomycetes community composition produced eigenvalues (λ) of 0.938, 0.674, 0.352 and 0.343, and GL 7.914, 3.049, 1.531 and 2.400 for the first four axes, respectively. We did not detect significant effects of the Plot variable nor of the woody debris variables tested. Finally, the DCA ordination for the Polypores community composition produced eigenvalues (λ) of 0.807, 0.694, 0.326 and 0.159 and GL of 8.991, 7.253, 2.570 and 1.262 for the first four axes, respectively. DCA plots

for Polyporoids and Heterobasidiomycetes are not shown since no significant variables were found.

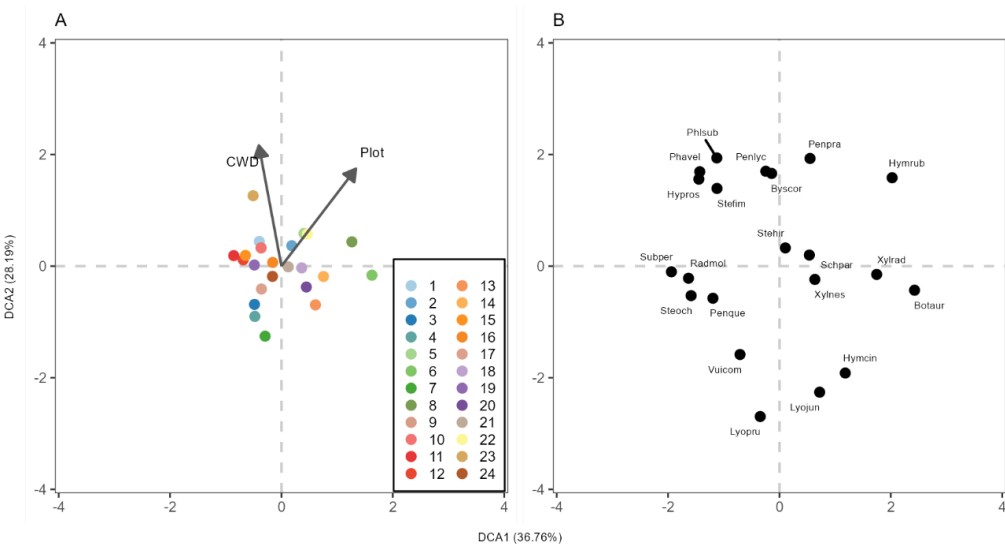

**Figure 5.** Detrended correspondence analysis (DCA) of Corticioid sporocarp community composition across plots. In (**A**), the studied plots are indicated with different dot colors while both Plot variable and the significant deadwood variables are indicated with arrows. In (**B**), the most abundant species are represented. The percentage of variance explained by each DCA axis is in parentheses. Abbreviations of fungal species shown in the DCA diagram: Botaur—*Botryobasidium aureum*, Byscor—*Byssomerulius corium*, Hymcin—*Hymenochaete cinnamomea*, Hymrub—*Hymenochaete rubiginosa*, Hypros—*Hyphoderma roseocremeum*, Lyojun—*Lyomyces juniperi*, Lyopru—*Lyomyces pruni*, Penlyc—*Peniophora lycii*, Penpra—*Peniophorella praetermissa*, Penque—*Peniophora quercina*, Phavel—*Phanerochaete velutina*, Phlsub—*Phlebia subochracea*, Radmol—*Radulomyces molaris*, Schpar—*Schizopora paradoxa*, Stefim—*Steccherinum fimbriatum*, Stehir—*Stereum hirsutum*, Steoch—*Steccherinum ochraceum*, Subper—*Subulicystidium perlongisporum*, Vuicom—*Vuilleminia comedens*, Xylnes—*Xylodon nesporii*, Xylrad—*Xylodon radula*.

## 4. Discussion

### 4.1. Wood-Decay Fungi Community Composition

We recorded a total of 1497 sporocarps comprising 156 species, with the fructification occurring principally during autumn. Spring, on the other hand, exhibited lower sporocarp values. This dissimilarity is expected because most of the groups studied have a preference to fructify during autumn when the meteorological conditions are favorable for sporocarp fructification [46]. In fact, specifically in Mediterranean oak woods, Salerni et al. [47] reported a maximum number of fruiting species in autumn since temperatures are mild and rains are abundant. Moreover, except for a few species exclusively detected in spring, most species encountered during this season were the same as those found in autumn. This similarity can be explained by two factors: (i) the majority of species can fructify throughout the year given suitable environmental conditions, and (ii) wood-decay fungi, specially Corticioids and Polyporoids, usually have wider fruiting longevity [24,48] and can be observed months after the fructification period. Furthermore, our study revealed the significant dominance of Corticioid fungi, corroborating the findings of Purhonen et al. [24] in *Picea abies* (L.) H.Karst. stands. A plausible explanation for this outcome is that Corticioid fungi can fructify on various woody debris features, as shown in Figure 1, and both its productivity and richness rise when increasing the size variety of the woody debris on the plots (Table 2). Consequently, it can be concluded that Corticioid fungi represent a paraphyletic group with a generalist nature yet comprising substrate-specialized fungi. This is in disagreement with the results of Granito et al. [49], which found corticolous fungi on CWD with cracked and loose bark (corresponding with DS2 in our study), being

pioneers and preparing the substrate for subsequent groups. In contrast, a lower number of species belonging to the Ascomycota group were detected, likely due to their reduced size, which makes them less conspicuous and more frequently overlooked, as reported by Purhonen et al. [24]. Additionally, Heterobasidiomycetes were less frequently listed in our study, possibly because their ephemeral sporocarps have a shorter persistence [24], reducing the likelihood of their detection during surveys.

Among the Corticioids and Polyporoids observed during this study, 31 species were found to be widely distributed in the Mediterranean basin [34,36,43]. In fact, a preliminary contribution on the macrofungi growing in deciduous oak (*Quercus cerris* L.) woods of the State Nature Reserves of Cornocchia and Palazzo, reported various species with a Mediterranean distribution, such as *Biscogniauxia mediterranea*, *Daedaleopsis nitida*, *Fuscoporia torulosa*, *Peniophora meridionalis*, *Postia simanii*, *Vitreoporus dichrous*, *Xenasmatella ardosiaca* and the genus *Hyphoderma* represented by three species (*H. medioburiense*, *H. nemorale* and *H. occidentale*) [50]. Notably, the identification of *Junghuhnia semisupiniformis* is noteworthy, as this species has thus far exclusively been detected in Italy [43], and *Subulicystidium perlongisporum*, which, conversely, would be a new record for Italy [36]. Also interesting is the finding of *Fuscopostia leucomallella*, a species typically associated with conifer substrates [43].

When plotting how the studied groups are distributed within the different woody types, we found that Corticioid, Ascomycota, Polyporoids and Heterobasidiomycetes did not show any preference for either the size or the decay stage of the woody debris. We expected this result, particularly for Coriticioid and Polyporoid funghi, since Saitta et al. [45] reported that these two groups colonize the wood of a very wide range of plants during all decay stages. For the other two groups studied, although they were found in all types of woody debris, higher fructification was detected in smaller sizes as already detected by Nordén et al. [13] for Ascomycetes.

As observed in the Venn diagrams, the first decay stage (DS1) hosted a higher number of species, while the most degraded woody debris (DS3) hosted the lowest values. Deepening, most of the hosted species of the first decay stage belongs to Ascomycota with genera such as *Hypoxylon*, and species such as *Diatrype stigma* and *Diderma radiatum*. Heterobasidiomycetes were also mostly found in the first decay stage, with genera such as *Auricularia* or *Tremella* detected exclusively in this decay class, and species such as *Exidiopsis calcea*, which were mostly found in the first decay stage, an outcome also reported by Küffer et al. [15]. Regarding the size of the woody debris, we observed that smaller sizes exhibited higher species richness, with very fine woody debris (VFWD) hosting most species, followed by fine woody debris (FWD), and coarse woody debris (CWD) having the lowest richness. This result is somewhat contrary to our expectations since bigger woody debris provides more resources in space and time for fungi than smaller woody debris [2,14]. One possible explanation for this result is that the smallest fraction studied (VFWD) experiences more variable moisture and temperature conditions compared to the other two fractions (FWD, CWD), thereby promoting the establishment of species with specific requirements and consequently increasing species richness. This highlights the importance of smaller fractions of woody debris in supporting the diversity of the wood-decay fungi, as reported previously [2,13], and having higher nutrient concentrations than wider woody debris sizes [51,52].

### 4.2. Wood-Decaying Fungi Productivity and Diversity and Plot-Specific Woody Size

When studying the effect of woody debris sizes found in the plots on the productivity and richness of wood-decay fungi, we observed a positive relationship between these variables. Moreover, the Variety variable had a positive effect in explaining the richness of the total Corticioid and Polyporoid species, indicating that a greater variety of woody debris led to a higher number of species, coinciding with Abrego and Salcedo [18] and Saitta et al. [45]. This is a logical result since different species have specific preferences for the size of the wood on which they grow [14], and a greater variety of woody debris allows

for a greater number of species. Ascomycetes, on the other hand, showed a preference for smaller fractions of woody debris, as already indicated by Nordén et al. [13], possibly because their sporocarps are smaller and lighter and can grow on smaller woody pieces containing less hardwood where heavier and larger species cannot fructify. As expected, we found that the number of Polyporoid fungi increased with the abundance of medium and large woody debris (i.e., FWD and CWD). This finding is in line with previous studies by Juutilainen et al. [14], indicating that Polyporoid fungi typically grow on large wood. This suggests that larger woody debris pieces provide the necessary support and resources for the growth of heavy and larger Polyporoid sporocarps. Regarding the productivity and richness of the Heterobasidiomycetes, we did not find any significant variables, most likely because the number of observations found was very low. With the obtained results, we saw that the specifical woody debris variables belonging to the different woody sizes (i.e., VFWD, FWD and CWD) and the variety of woody debris have a significant effect when explaining how both the productivity and richness of wood-decay fungi are modeled. On the other hand, the total amount of woody debris (Total_WD in Table 2) did not show a significant effect on the variables tested, possibly indicating that the heterogeneity of the woody debris could be more determinant for the wood-inhabiting fungi fructification than the total amount of woody debris.

### 4.3. Wood-Decaying Fungi Community Composition and Plot-Specific Woody Size

When analyzing the sporocarps' relative abundance and community composition in the studied area, we found that among the specific-plot woody debris variables, only FWD and CWD had a significant effect on the total community composition as well as on the subgroups studied, including Ascomycota and Corticioids. However, for the Polypores and Heterobasidiomycetes, none of the variables tested had a significant effect on the community composition. This lack of significance may be attributed to the low number of sporocarps found for these groups in our study. Interestingly, while the Plot variable itself was not significant for the studied groups, we observed that it was the primary factor driving the wood-decay fungal community composition. This finding suggests that the differences in the fungal communities are primarily dependent on the specific characteristics of each individual plot. Local factors such as morphology, topography, and microclimate within the plots may have a more substantial influence on the fungal communities than the overall characteristics of the studied area [53]. This supports the idea that wood-decay fungal communities exhibit high spatial turnover among logs within a forest [18,54]. In other words, the composition of fungal communities can vary significantly at small spatial scales due to localized environmental conditions and microhabitat differences.

### 5. Conclusions

Our study provides valuable insights into the fructification patterns and dominance of wood-decay fungal groups in Mediterranean deciduous oak forests. By examining the distribution patterns across different woody debris, we gained insights into the ecological preferences and responses of fungal species to decay stages and size fractions. However, it is crucial to acknowledge that the turnover and spatial heterogeneity of wood-decay fungal communities may be influenced by unmeasured factors such as specific tree species, microhabitats, and interactions with other organisms. Future research should explore these additional factors to obtain a more comprehensive understanding of the dynamics and drivers of wood-decay fungal communities in the studied area. Furthermore, the utilization of alternative methodologies such as DNA metabarcoding can assist in obtaining a more comprehensive understanding of the fungal community, mitigating potential biases that may arise from solely relying on sporocarp surveys. By doing so, we can further elucidate the ecological roles and interactions of these fungi in the decomposition processes of woody debris.

**Author Contributions:** Conceptualization, M.N.D.; Methodology, Á.P. and M.N.D.; Validation, E.S. and C.P.; Formal Analysis, Á.P.; Investigation, Á.P., E.S., M.N.D. and C.P.; Data Curation, Á.P., E.S. and C.P.; Writing—Original Draft Preparation, Á.P.; Writing—Review and Editing, Á.P., E.S., M.N.D. and C.P.; Visualization, Á.P., E.S. and C.P.; Supervision, E.S.; Project Administration, C.P.; Funding Acquisition, C.P. All authors have read and agreed to the published version of the manuscript.

**Funding:** This work was supported by the Secretariat for Universities and of the Ministry of Business and Knowledge of the Government of Catalonia and the European Social Fund and the Spanish Ministry of Science, Innovation and Universities, grant RTI2018-099315-A-I00. The authors acknowledge the support of NBFC to the University of Siena, funded by the Italian Ministry of University and Research, PNRR, Missione 4, Componente 2, "Dalla ricerca all'impresa", Investimento 1.4, Project CN00000033.

**Data Availability Statement:** The data presented in this study are available on request from the corresponding author.

**Acknowledgments:** We thank the "Corpo Forestale dello Stato, Ufficio Territoriale per la Biodiversità—Siena", for their willingness and logistic support, especially Carlo Saveri and Marco Landi. Our gratitude also to Pamela Leonardi and Diego Cantini for their precious help in fieldwork and laboratory. Finally, we would also like to thank Albert Morera for his great assistance in designing the figures included in this study.

**Conflicts of Interest:** The authors declare no conflict of interest.

## Appendix A

**Table A1.** List of the species found in the State Nature Reserves of Cornocchia (COR) and Palazzo (PAL) with information belonging to the season in which they were surveyed, de decay stage (DS1, 1; DS2; 2 and DS3, 3) and the size (VFWD, V; FWD, F; CWD, C) of the wood and the studied group they belong (Ascomycetes, A; Corticioids, C; Heterobasidiomycetes, H; Polyporoids, P).

| Species | Study Area | Season | Decay | Size | Group |
|---|---|---|---|---|---|
| *Abortiporus biennis* (Bull.) Singer | PAL | Autumn | 1 | C | P |
| *Amylostereum laevigatum* (Fr.) Boidin | COR/PAL | Spring/Autumn | 1, 2 | V | C |
| *Antrodia albida* (Fr.) Donk | PAL | Autumn | 1 | V | P |
| *Antrodia ramentacea* (Berk. & Broome) Donk | COR/PAL | Spring/Autumn | 1, 3 | V, F | P |
| *Antrodiella genistae* (Bourdot & Galzin) A. David | PAL | Autumn | 1 | V | P |
| *Antrodiella romellii* (Donk) Niemelä | COR/PAL | Spring/Autumn | 1, 3 | V, F | P |
| *Antrodiella semisupina* (Berk. & M.A. Curtis) Ryvarden | COR/PAL | Autumn | 1 | V, C | P |
| *Athelia decipiens* (Höhn. & Litsch.) J. Erikss. | COR/PAL | Autumn | 1 | V | C |
| *Athelia epiphylla* Pers. | COR/PAL | Spring/Autumn | 1, 3 | V, F | C |
| *Athelia* sp. | COR | Autumn | 5 | F | C |
| *Athelia* sp. 2 | PAL | Autumn | 1 | V | C |
| *Athelia tenuispora* Jülich | PAL | Autumn | 3 | F | C |
| *Atheliachaete galactites* (Bourdot & Galzin) Ţura, Zmitr., Wasser & Spirin | COR/PAL | Autumn | 1, 3 | V | C |
| *Atheliachaete sanguinea* (Fr.) Spirin & Zmitr. | PAL | Autumn | 1 | V | C |
| *Auricularia auricula-judae* (Bull.) Quél. | COR/PAL | Autumn | 1 | V, F, C | H |
| *Auricularia mesenterica* (Dicks.) Pers. | COR/PAL | Spring/Autumn | 1 | V, F, C | H |
| *Basidiodendron caesiocinereum* (Höhn. & Litsch.) Luck-Allen | COR/PAL | Autumn | 1, 3 | V, C | H |
| *Biscogniauxia mediterranea* (De Not.) Kuntze | PAL | Autumn | 1 | F | A |
| *Bjerkandera adusta* (*Willd.*) P. Karst. | PAL | Autumn | 2 | V | P |
| *Botryobasidium aureum* Parmasto | PAL | Spring/Autumn | 1, 2, 3 | V, F, C | C |
| *Botryobasidium laeve* (J. Erikss.) Parmasto | COR/PAL | Spring/Autumn | 1, 2, 3 | V, F, C | C |
| *Botryohypochnus isabellinus* (Fr.) J. Erikss. | COR/PAL | Autumn | 1, 2 | V | C |
| *Byssomerulius corium* (Pers.) Parmasto | COR/PAL | Autumn | 1, 3 | V, F | C |
| *Byssomerulius hirtellus* (Burt) Parmasto | COR | Spring/Autumn | 2, 3 | V, F | C |
| *Calocera cornea* (Batsch) Fr | COR/PAL | Autumn | 1, 3 | V | H |
| *Calycina citrina* (Hedw.) Gray | COR/PAL | Autumn | 1, 2 | V | A |
| *Capitotricha bicolor* (Bull.) Baral | PAL | Spring | 1 | V | A |
| *Ceriporia excelsa* S. Lundell ex Parmasto | COR | Autumn | 3 | V | P |
| *Ceriporia purpurea* (Fr.) Donk | COR/PAL | Spring | 2, 3 | V | P |
| *Ceriporiopsis mucida* (Pers.) Gilb. & Ryvarden | COR/PAL | Spring/Autumn | 1, 2, 3 | V, C | P |

**Table A1.** *Cont.*

| Species | Study Area | Season | Decay | Size | Group |
|---|---|---|---|---|---|
| *Cyanosporus subcaesius* (A. David) B.K. Cui, L.L. Shen & Y.C. Dai | COR/PAL | Spring/Autumn | 1, 2, 3 | V, F | P |
| *Cylindrobasidium laeve* (Pers.) Chamuris | COR/PAL | Spring/Autumn | 1, 2 | V, F, C | C |
| *Dacrymyces stillatus* Nees | COR/PAL | Autumn | 1 | V | H |
| *Daedaleopsis nitida* (Durieu & Mont.) Zmitr. & Malysheva | COR/PAL | Spring/Autumn | 1, 3 | V, F, C | P |
| *Dasyscyphella nivea* (R. Hedw.) Raitv. | COR/PAL | Spring/Autumn | 1, 3 | V, F | A |
| *Dendrothele acerina* (Pers.) P.A. Lemke | PAL | Autumn | 1, 3 | V | C |
| *Diatrype stigma* (Hoffm.) Fr. | PAL | Spring/Autumn | 1 | V | A |
| *Diderma radiatum* (L.) Morgan | COR/PAL | Spring | 1 | V | A |
| *Ditiola peziziformis* (Lév.) D.A. Reid | PAL | Autumn | 2 | V | A |
| *Efibula tuberculata* (P. Karst.) Zmitr. & Spirin | COR/PAL | Autumn | 1, 3 | V | C |
| *Eutypa scabrosa* (Bull.) Auersw. | COR | Autumn | 1 | V | A |
| *Exidia glandulosa* (Bull.) Fr. | COR/PAL | Spring/Autumn | 1, 2, 3 | V, F, C | H |
| *Exidia recisa* (Ditmar) Fr. | PAL | Autumn | 1 | F | H |
| *Exidia thuretiana* (Lév.) Fr. | COR/PAL | Autumn | 1 | V | H |
| *Exidiopsis calcea* (Pers.) K. Wells | COR/PAL | Spring/Autumn | 1, 3 | V, F | H |
| *Exidiopsis effusa* Bref. | PAL | Autumn | 1 | V | H |
| *Fasciodontia bugellensis* (Ces.) Yurchenko, Riebesehl & Langer | PAL | Autumn | 1 | V | C |
| *Fibrodontia gossypina* Parmasto | COR/PAL | Autumn | 3 | V | C |
| *Fomitiporia robusta* (P. Karst.) Fiasson & Niemelä | COR | Autumn | 1 | V, F | P |
| *Fuscoporia contigua* (Pers.) G. Cunn. | COR | Autumn | 1 | F | P |
| *Fuscoporia ferruginosa* (Schrad.) Murrill | COR/PAL | Spring/Autumn | 3 | V, F | P |
| *Fuscoporia torulosa* (Pers.) T. Wagner & M. Fisch. | COR/PAL | Autumn | 1 | V, F | P |
| *Fuscopostia leucomallella* (Murrill) B.K. Cui, L.L. Shen & Y.C. Dai | PAL | Autumn | 1, 3 | V, C | P |
| *Ganoderma australe* (Fr.) Pat. | PAL | Autumn | 1 | C | P |
| *Hapalopilus rutilans* (Pers.) Murrill | COR/PAL | Spring/Autumn | 1, 3 | V, F, C | P |
| *Heteroradulum deglubens* (Berk. & Broome) Spirin & Malysheva | COR | Spring/Autumn | 1 | V | H |
| *Hymenochaete cinnamomea* (Pers.) Bres | COR/PAL | Spring/Autumn | 1, 2, 3 | V, F, C | C |
| *Hymenochaete rubiginosa* (Dicks.) Lév. | COR/PAL | Spring/Autumn | 1 | V, F, C | C |
| *Hymenoscyphus calyculus* (Fr.) W. Phillips | PAL | Spring | 1 | V | A |
| *Hyphoderma crustulinum* (Bres.) Nakasone | PAL | Spring | 3 | F | C |
| *Hyphoderma litschaueri* (Burt) J. Erikss. & Å. Strid | COR | Autumn | 1, 2 | V | C |
| *Hyphoderma medioburiense* (Burt) Donk | PAL | Autumn | 1 | V | C |
| *Hyphoderma nemorale* K.H. Larss. | COR/PAL | Spring/Autumn | 1, 3 | V, F, C | C |
| *Hyphoderma occidentale* (D.P. Rogers) Boidin & Gilles | COR/PAL | Spring/Autumn | 1, 3 | V, F | C |
| *Hyphoderma orphanellum* (Bourdot & Galzin) Donk | COR | Autumn | 1 | V | C |
| *Hyphoderma roseocremeum* (Bres.) Donk | COR/PAL | Spring/Autumn | 1, 2, 3 | V, F, C | C |
| *Hyphoderma setigerum* (Fr.) Donk | COR/PAL | Spring/Autumn | 1, 2 | V, F | C |
| *Hyphoderma* sp. | PAL | Autumn | 3 | V | C |
| *Hyphoderma transiens* (Bres.) Parmasto | COR | Spring | 3 | F | C |
| *Hyphodontia alutaria* (Burt) J. Erikss. | COR | Autumn | 1 | V | C |
| *Hyphodontia arguta* (Fr.) J. Erikss. | COR | Autumn | 1 | F | C |
| *Hyphodontia quercina* (Pers.) J. Erikss. | COR/PAL | Spring/Autumn | 1, 3 | V, F | C |
| *Hypochnicium cremicolor* (Bres.) H. Nilsson & Hallenb. | COR | Spring | 3 | C | C |
| *Hypoxylon fuscum* (Pers.) Fr. | COR/PAL | Spring/Autumn | 1 | V | A |
| *Hypoxylon rubiginosum* (Pers.) Fr. | COR/PAL | Spring/Autumn | 1, 2 | V, F, C | A |
| *Incrustoporia chrysella* (Niemelä) Zmitr. | PAL | Autumn | 3 | F | P |
| *Irpex lacteus* (Fr.) Fr. | COR | Autumn | 1 | C | P |
| *Junghuhnia nitida* (Pers.) Ryvarden | COR/PAL | Spring/Autumn | 1, 2, 3 | V, F, C | P |
| *Lachnum virgineum* (Batsch) P. Karst. | COR/PAL | Spring | 1 | V | A |
| *Laxitextum bicolor* (Pers.) Lentz | COR | Autumn | 1 | V | P |
| *Lindtneria chordulata* (D.P. Rogers) Hjortstam | COR/PAL | Autumn | 1 | V | C |
| *Lyomyces crustosus* (Pers.) P. Karst. | COR/PAL | Spring/Autumn | 1, 3 | V, F | C |
| *Lyomyces juniperi* (Bourdot & Galzin) Riebesehl & Langer | COR/PAL | Spring/Autumn | 1, 3 | V | C |
| *Lyomyces pruni* (Lasch) Riebesehl & Langer | COR/PAL | Spring/Autumn | 1, 2 | V, F | C |
| *Lyomyces sambuci* (Pers.) P. Karst. | PAL | Autumn | 3 | V | C |
| *Mycoacia aurea* (Fr.) J. Erikss. & Ryvarden | COR | Spring | 3 | F | C |
| *Mycoacia fuscoatra* (Fr.) Donk | PAL | Spring | 1 | V | C |
| *Mycoacia livida* (Pers.) Zmitr. | COR | Spring/Autumn | 2, 3 | V, F | C |
| *Mycoacia nothofagi* (G. Cunn.) Ryvarden | COR | Spring | 3 | V | C |
| *Mycoacia uda* (Fr.) Donk | COR | Autumn | 1 | V | C |
| *Mycoaciella bispora* (Stalpers) J. Erikss. & Ryvarden | COR | Autumn | 1 | V | C |

**Table A1.** *Cont.*

| Species | Study Area | Season | Decay | Size | Group |
|---|---|---|---|---|---|
| *Neoantrodia serialis* (Fr.) Audet | COR/PAL | Spring/Autumn | 2, 3 | V, C | P |
| *Oligoporus* sp. | COR | Autumn | 3 | F | P |
| *Orbilia coccinella* Fr. | COR | Autumn | 3 | V | A |
| *Orbilia xanthostigma* (Fr.) Fr. | COR/PAL | Autumn | 1, 2 | V, F | A |
| *Peniophora boidinii* D.A. Reid | PAL | Spring | 1 | V | C |
| *Peniophora cinerea* (Pers.) Cooke | COR/PAL | Autumn | 1, 3 | V, C | C |
| *Peniophora incarnata* (Pers.) P. Karst. | COR | Autumn | 1 | V | C |
| *Peniophora lycii* (Pers.) Höhn. & Litsch. | COR/PAL | Spring/Autumn | 1 | V | C |
| *Peniophora meridionalis* Boidin | COR/PAL | Autumn | 1 | V | C |
| *Peniophora quercina* (Pers.) Cooke | COR/PAL | Spring/Autumn | 1, 2 | V, F, C | C |
| *Peniophorella praetermissa* (P. Karst.) K.H. Larss. | COR/PAL | Spring/Autumn | 1, 2, 3 | V, F, C | C |
| *Phaeophlebiopsis ravenelii* (Cooke) Zmitr. | COR/PAL | Autumn | 1, 2, 3 | V, F | C |
| *Phanerochaete calotricha* (P. Karst.) J. Erikss. & Ryvarden | PAL | Autumn | 1 | F | C |
| *Phanerochaete laevis* (Fr.) J. Erikss. & Ryvarden | COR | Spring | 1 | V | C |
| *Phanerochaete sordida* (P. Karst.) J. Erikss. & Ryvarden | PAL | Spring/Autumn | 1, 3 | V | C |
| *Phanerochaete velutina* (DC.) P. Karst. | COR/PAL | Spring/Autumn | 1, 3 | V, F, C | C |
| *Phellinus pomaceus* (Pers.) Maire | COR | Spring | 1 | V, F | P |
| *Phlebia lilascens* (Bourdot) J. Erikss. & Hjortstam | PAL | Autumn | 2 | V | C |
| *Phlebia* sp. | COR/PAL | Autumn | 1 | V, F | C |
| *Phlebia subochracea* (Alb. & Schwein.) J. Erikss. & Ryvarden | COR/PAL | Spring | 1, 3 | V, F | C |
| *Postia simanii* (Pilát ex Pilát) Jülich | PAL | Spring | 3 | F | P |
| *Propolis farinosa* (Pers.) Fr. | COR | Autumn | 1, 2 | V | A |
| *Radulomyces confluens* (Fr.) M.P. Christ. | PAL | Autumn | 1 | V | C |
| *Radulomyces molaris* (Chaillet ex Fr.) M.P. Christ. | COR/PAL | Spring/Autumn | 1, 2, 3 | V, F, C | C |
| *Resiniporus resinascens* (Romell) Zmitr. | COR | Autumn | 3 | F | P |
| *Rutstroemia bolaris* (Batsch) Rehm | PAL | Autumn | 1 | V | A |
| *Rutstroemia firma* (Pers.) P. Karst. | PAL | Autumn | 1 | V | A |
| *Sarcoscypha coccinea* (Gray) Boud. | COR/PAL | Autumn | 1, 3 | V | A |
| *Schizopora paradoxa* (Schrad.) Donk | COR/PAL | Spring/Autumn | 1, 2, 3 | V, F, C | C |
| *Scutellinia kerguelensis* (Berk.) Kuntze | COR | Spring | 1, 2, 3 | V, C | A |
| *Sebacina* sp. | COR | Autumn | 1 | V | H |
| *Sidera vulgaris* (Fr.) Miettinen | COR/PAL | Spring/Autumn | 2, 3 | F, C | P |
| *Skeletocutis nivea* (Jungh.) Jean Keller | COR/PAL | Spring/Autumn | 1, 2, 3 | V, F, C | P |
| *Skeletocutis percandida* (Malençon & Bertault) Jean Keller | COR | Autumn | 3 | F | P |
| *Steccherinum fimbriatum* (Pers.) J. Erikss. | COR/PAL | Spring/Autumn | 1, 3 | V, F, C | C |
| *Steccherinum lacerum* (P. Karst.) Kotir. & Saaren. | COR | Spring/Autumn | 1 | F | C |
| *Steccherinum ochraceum* (Pers.) Gray | COR/PAL | Spring/Autumn | 1, 2, 3 | V, F, C | C |
| *Steccherinum semisupiniforme* (Murrill) Miettinen | PAL | Autumn | 1 | V | P |
| *Stereum gausapatum* (Fr.) Fr. | COR/PAL | Spring/Autumn | 1 | V, F | C |
| *Stereum hirsutum* (Willd.) Pers. | COR/PAL | Spring/Autumn | 1, 2, 3 | V, F, C | C |
| *Stereum ochraceoflavum* (Schwein.) Sacc. | COR/PAL | Spring/Autumn | 1 | V, F | C |
| *Stereum reflexulum* Lloyd | COR | Autumn | 1 | V | C |
| *Subulicystidium longisporum* (Pat.) Parmasto, | PAL | Autumn | 1, 2 | V, C | C |
| *Subulicystidium perlongisporum* Boidin & Gilles | COR/PAL | Spring/Autumn | 1, 2, 3 | V, F, C | C |
| *Szczepkamyces campestris* (Quél.) Zmitr. | COR/PAL | Autumn | 1 | V, F | P |
| *Tapesia fusca* (Pers.) Fuckel | PAL | Autumn | 1 | V | A |
| *Terana coerulea* (Lam.) Kuntze | COR/PAL | Autumn | 1 | V | C |
| *Tomentella asperula* (P. Karst.) Höhn. & Litsch. | COR | Autumn | 1 | C | C |
| *Tomentella ferruginea* (Pers.) Pat. | PAL | Autumn | 3 | F | C |
| *Trametes ochracea* (Pers.) Gilb. & Ryvarden | PAL | Autumn | 1, 3 | F, C | P |
| *Trametes versicolor* (L.) Lloyd | COR | Autumn | 1 | V | P |
| *Trechispora cohaerens* (Schwein.) Jülich & Stalpers | PAL | Autumn | 1 | V | C |
| *Trechispora farinacea* (Pers.) Liberta | COR | Autumn | 1, 3 | V | C |
| *Trechispora fastidiosa* (Pers.) Liberta | COR | Autumn | 3 | F | C |
| *Trechispora microspora* (P. Karst.) Liberta | PAL | Autumn | 1 | V | C |
| *Trechispora mollusca* (Pers.) Liberta | COR/PAL | Spring/Autumn | 2, 3 | V, F, C | C |
| *Trechispora nivea* (Pers.) K.H. Larss. | COR | Spring | 3 | F | C |
| *Tremella globispora* D.A. Reid | PAL | Autumn | 1 | V | H |
| *Tremella mesenterica* Retz. | COR | Autumn | 1 | V | H |
| *Trichaptum biforme* (Fr.) Ryvarden | COR/PAL | Spring/Autumn | 1, 2, 3 | F, C | P |
| *Tubulicrinis medius* (Bourdot & Galzin) Oberw. | COR | Autumn | 1 | F | C |
| *Tulasnella pallida* Bres. | COR/PAL | Autumn | 1, 3 | V, F, C | H |
| *Vitreoporus dichrous* (Fr.) Zmitr. | COR | Autumn | 1 | V, C | P |
| *Vuilleminia comedens* (Nees) Maire | COR/PAL | Spring/Autumn | 1, 3, 5 | V, F | C |
| *Xenasmatella ardosiaca* (Bourdot & Galzin) Stalpers | COR | Autumn | 1, 3 | V, C | C |

**Table A1.** *Cont.*

| Species | Study Area | Season | Decay | Size | Group |
|---|---|---|---|---|---|
| *Xylaria hypoxylon* (L.) Grev. | COR/PAL | Spring/Autumn | 1, 2, 3 | V, F, C | A |
| *Xylodon asper (Fr.)* Hjortstam & Ryvarden | COR | Autumn | 1, 3 | V, F | C |
| *Xylodon brevisetus* (P. Karst.) Hjortstam & Ryvarden | COR/PAL | Spring/Autumn | 1, 2, 3 | V, F | C |
| *Xylodon flaviporus* (Berk. & M.A. Curtis ex Cooke) Riebesehl & Langer | COR/PAL | Spring/Autumn | 1, 2, 3 | F, C | C |
| *Xylodon nesporii* (Bres.) Hjortstam & Ryvarden | COR/PAL | Spring/Autumn | 1, 2, 3 | V, F | C |
| *Xylodon radula* (Fr.) Ţura, Zmitr., Wasser & Spirin (Fr.) Nobles | COR/PAL | Autumn | 1, 2, 3 | V, F, C | C |
| *Xylodon raduloides* Riebesehl & Langer | COR/PAL | Spring/Autumn | 1, 3 | V, F | C |

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
