# Peer review of "Wood-Decay Fungi Fructifying in Mediterranean Deciduous Oak Forests: A Community Composition, Richness and Productivity Study"

_forests, doi:10.3390/f14071326_

Round 1
Reviewer 1 Report
In general, this is a well-written manuscript
The topic of the work sounds interesting.
Some English language editorial changes are needed in the text
- check carefully the formats, especially concerning citations and references
In addition, it would be great to provide host plants/habitat for fungal species table Appendix A
Some English language editorial changes are needed in the text
Reviewer 2 Report
As a reviewer of the article titled "Wood-decay fungi fructifying in Mediterranean deciduous oak forests: a community composition, richness and productivity study," I have evaluated the content and provide the following feedback.
Overall, the article is suitable for publication; however, there are some areas that require attention. The introduction section needs to be shortened and the language revised to improve clarity. Additionally, there are grammatical errors that should be corrected.
Regarding Figure 2 and Figure 4, further explanations are needed to enhance their understanding. Please provide more detailed descriptions or captions for these figures to ensure that readers can fully comprehend the information presented.
In general, the article provides valuable insights into the colonization patterns of wood-decay fungi in Mediterranean Quercus Cerris forests. The study examines the influence of decay stage and size of deadwood on the community composition, richness, and productivity of wood-decay fungi. The results highlight the differential responses of Ascomycetes, Corticioids, Polyporoids, and Heterobasidiomycetes to various woody debris classes. Moreover, the study emphasizes the significance of smaller and soft-decayed woody debris in supporting species richness and the positive effect of woody debris size heterogeneity on wood-decay fungi productivity.
In conclusion, with the necessary revisions, this article can make a valuable contribution to the field of wood science and forest products. Please address the mentioned points and resubmit the revised version for further consideration.
As a reviewer of the article titled "Wood-decay fungi fructifying in Mediterranean deciduous oak forests: a community composition, richness and productivity study," I have evaluated the content and provide the following feedback.
Overall, the article is suitable for publication; however, there are some areas that require attention. The introduction section needs to be shortened and the language revised to improve clarity. Additionally, there are grammatical errors that should be corrected.
Regarding Figure 2 and Figure 4, further explanations are needed to enhance their understanding. Please provide more detailed descriptions or captions for these figures to ensure that readers can fully comprehend the information presented.
In general, the article provides valuable insights into the colonization patterns of wood-decay fungi in Mediterranean Quercus Cerris forests. The study examines the influence of decay stage and size of deadwood on the community composition, richness, and productivity of wood-decay fungi. The results highlight the differential responses of Ascomycetes, Corticioids, Polyporoids, and Heterobasidiomycetes to various woody debris classes. Moreover, the study emphasizes the significance of smaller and soft-decayed woody debris in supporting species richness and the positive effect of woody debris size heterogeneity on wood-decay fungi productivity.
In conclusion, with the necessary revisions, this article can make a valuable contribution to the field of wood science and forest products. Please address the mentioned points and resubmit the revised version for further consideration.
